



# Ozone pollution in China affected by stratospheric quasi-biennial oscillation

Mengyun Li[1], Yang Yang[1*], Hailong Wang[2], Huimin Li[1], Pinya Wang[1], Hong Liao[1]

[1]Jiangsu Key Laboratory of Atmospheric Environment Monitoring and Pollution Control, Jiangsu Collaborative Innovation Center of Atmospheric Environment and Equipment Technology, School of Environmental Science and Engineering, Nanjing University of Information Science and Technology, Nanjing, Jiangsu, China

[2]Atmospheric Sciences and Global Change Division, Pacific Northwest National Laboratory, Richland, Washington, USA

*Correspondence to yang.yang@nuist.edu.cn





## Abstract

In recent years, near-surface ozone ($O_3$) level has been rising fast in China,
with increasing damages to human health and ecosystems. In this study, the
impact of stratospheric quasi-biennial oscillation (QBO) on interannual
variations in summertime tropospheric $O_3$ over China is investigated based on
GEOS-Chem model simulations and satellite retrievals. QBO has a significant
positive correlation with near-surface $O_3$ concentrations over central China
(92.5°–112.5°E, 26°–38°N) when the sea surface temperature (SST) over the
eastern tropical Pacific is warmer than normal, with a correlation coefficient of
0.53, but QBO has no significant effect on $O_3$ under the cold SST anomaly.
Compared to the easterly phase of QBO, the near-surface $O_3$ concentrations
have an increase of up to 3 ppb (5% relative to the average) over central China
during its westerly phase under the warm SST anomaly. $O_3$ also increases
above the surface and up to the upper troposphere, with a maximum increase
of 2–3 ppb (3–5%) in 850–500 hPa over central China comparing westerly
phase to easterly phase. Process-based analysis and sensitivity simulations
suggest that the $O_3$ increase over central China is mainly attributed to the
anomalous downward transport of $O_3$ during the westerly phase of QBO when
a warm SST anomaly occurs in the eastern tropical Pacific, while the local
chemical reactions and horizontal transport processes partly offset the $O_3$
increase. This work suggests a potentially important role of QBO and the
related vertical transport process in affecting near-surface $O_3$ air quality, with
an indication for $O_3$ pollution prediction and prevention.





## 1. Introduction

Ozone ($O_3$) is an important atmospheric trace gas. The presence of $O_3$ in the stratosphere plays a crucial role in protecting the environment and humans from UV light, but it is detrimental to human health, ecosystem and agricultural production within the troposphere (Wang et al., 2007; Nuvolone et al., 2018; Zhao et al., 2020). Tropospheric $O_3$ is primarily produced by photochemical reactions of nitrogen oxides ($NO_x$) and volatile organic compounds (VOCs) (Wang et al., 2017). Apart from precursor emissions, the temporal and spatial distribution of tropospheric $O_3$ is highly impacted by meteorological conditions, among which low relative humidity (RH), cloud free, strong solar radiation and high temperatures can lead to $O_3$ pollution by enhancing its chemical production (Camalier et al., 2007; Porter et al., 2019; Gong and Liao, 2019; Qu et al., 2021; Wang et al., 2022). The downward transport of stratospheric $O_3$ into the troposphere is also one of the sources for near-surface $O_3$ (Zeng et. al., 2010; Wespes et al., 2017).

With the accelerated industrialization and urbanization in recent decades, the air quality problem has become serious in China (Verstraeten et al., 2015; Yang et al., 2022). Although many environmental protection and control measures have been implemented to prevent air pollution (Feng et al., 2019), $O_3$ pollution is getting worse in China in the most recent decade (Li et al., 2019; Gao et al., 2022). Therefore, the factors causing $O_3$ pollution have been a research focus in recent years. Many previous studies found that interannual variations in large-scale circulations modulated the $O_3$ pollution in China (e.g., Yang et al., 2014; Yin et al., 2017; Zhao and Wang, 2017). For example, Yang et al. (2014) showed that summertime $O_3$ levels in China were positively correlated with the strength of East Asian summer monsoon (EASM) associated with variations in large-scale circulations, which led to an increase in $O_3$ concentration exceeding 6% in the strong EASM years relative to the weak ones.



El Niño–Southern Oscillation (ENSO) is the most pronounced mode of internal variability of the Earth's climate system, which contains warm (El Niño) and cold (La Niña) phases, describing anomalous warming and cooling of surface waters, respectively, in the central-eastern tropical Pacific Ocean. It can be responsible for global and regional oceanic and atmospheric pattern anomalies, having significant impacts on wind, temperature and precipitation in China (Zhou et at., 2007; Xu et al., 2007; Li et al., 2020; Zeng et al., 2021). Recent studies have shown that the interannual variations in $O_3$ concentrations over China are influenced by ENSO events (Jiang and Li, 2022; Li et al., 2022; Yang et al., 2022). Using the GEOS-Chem model, Yang et al. (2022) showed that summertime $O_3$ over southern China had a positive correlation with Niño 3.4 index. Near-surface $O_3$ concentrations increased by a maximum over 20% in southern China during El Niño compared to La Niña, which was closely linked to a weakened southerly over southern China that was conducive to the accumulation of $O_3$. However, they also reported an unusual $O_3$ changes over 30°–40°N in China that could not be explained by the ENSO impact alone.

Quasi-Biennial Oscillation (QBO) is a major mode of variability of zonal wind in the stratosphere characterized by alternating easterly and westerly, with a period close to 28 months. The QBO is able to modulate large-scale vertical and meridional circulations between tropics and subtropics (Punge et al., 2009), which impacts the East Asian climate. For example, it is reported that southern China sea summer monsoon is weakened during the QBO westerly phase due to the associated anomalous Hadley-like circulation (Zheng et al., 2007). Kim et al. (2020) revealed that there was stronger precipitation over East Asia with a larger Madden-Julian oscillation (MJO) amplitude when the QBO is in easterly phase, which is because MJO-induced vertical motion and moisture transport is amplified by easterly QBO. Therefore, it is of interest to explore the influence of QBO on interannual variations in summertime $O_3$ pollution over China and their connections during anomalous modes of sea surface temperature (SST)





over the eastern tropical Pacific, as well as the mechanisms involved.
In the present study, the impact of QBO on $O_3$ variations in China is
examined based on GEOS-Chem simulations over 1981–2020, together with
satellite retrievals. The paper is organized as follows: In Section 2, we describe
the model, numerical experiments, the reanalysis datasets, the indices used in
this study and satellite retrieval data. The connection between QBO and
tropospheric $O_3$ in China and the possible mechanisms are explored in Section
3. Conclusions and discussion are given in Section 4.
**2. Methods**
**2.1. GEOS-Chem model simulations**
GEOS-Chem is a global three-dimensional chemical transport model with
comprehensive chemistry mechanisms of $O_3$-NOx-hydrocarbon-aerosol
involved in the model (Zhai et al., 2021). In this study, we apply the GEOS-
Chem version 12.9.3 to simulate $O_3$ from 1981 to 2020. The horizontal grid of
the model is 2° × 2.5° (latitude × longitude) with 47 vertical levels above the
surface. Stratospheric $O_3$ is calculated based on the linearized chemistry
mechanism (McLinden et al., 2000). Meteorological fields driving the GEOS-
Chem simulations are from the Modern-Era Retrospective analysis for
Research and Applications version 2 (MERRA-2) (Gelaro et al., 2017),
produced by the NASA's Global Modeling and Assimilation Office.
Anthropogenic emissions in China are obtained from the Multi-resolution
Emission Inventory for China (MEIC) (Zheng et al., 2018). Anthropogenic
emissions outside China are adopted from the Community Emissions Data
System (CEDS) version 20210205 (Hoesly et al., 2018). Biogenic emissions
employ the Model of Emissions of Gases and Aerosols from Nature (MEGAN)
version 2.1 (Guenther et al., 2012). Global biomass burning emissions are from
the Global Fire Emissions Database version 4 (GFED4) (van der Werf et al.,
2017). Soil $NO_x$ emissions are estimated in a soil parameterization scheme
(Hudman et al., 2012). Lightning-produced $NO_x$ emissions are estimated in the





model based on Murray et al. (2012) and Ott et al. (2010).

The GEOS-Chem simulations are performed to assess the impact of QBO

on interannual variation of $O_3$ covering the period of 1981–2020, following a 6-
month model spin-up. In order to minimize the impact of interannual variations
in emissions on the modeled $O_3$ concentrations, the anthropogenic, biogenic
burning and natural emissions of $O_3$ precursors are all fixed at their 2017 levels
in the base simulation (BASE). The BASE simulation is analyzed to quantify the
impact of QBO on $O_3$, unless stated otherwise.

A sensitivity simulation (NO_CHN) is conducted with a different emission

configuration than BASE, aiming to investigate the impact of domestic
emissions in China on tropospheric $O_3$ during QBO events. Different from BASE,
anthropogenic emissions of $NO_x$, CO and VOCs in China are turned off in
NO_CHN. Considering that $O_3$ pollution is most critical during the boreal
summer, only summer months (June-July-August, JJA) are examined in this
study. Time-varying meteorological fields follow those from MERRA-2 during all
simulations.

Figure 1 compares the year-by-year changes in JJA $O_3$ concentrations in

observations and BASE simulation. GEOS-Chem can roughly capture the
interannual variation in surface $O_3$ concentrations in China during 2016–2020.
The spatial correlation coefficients between the observed and modeled year-
by-year changes in $O_3$ concentrations are about 0.5–0.6, except the 2018-to-
2019 changes in $O_3$, which can be attributed to the strong influence of
emissions on observed $O_3$ concentrations.
**2.2. QBO and Niño 3.4 indices**

The QBO phases are determined by the zonal average of 30 hPa zonal

wind over the equator (5°S–5°N) based on MERRA-2 reanalysis (Fig. 2a).
Positive values denote westerly QBO phase (QBOW), while negative values
denote easterly QBO phase (QBOE).

Niño 3.4 index is used to characterize the warm and cold phases of SST





anomaly over the eastern tropical Pacific, which is estimated as the SST
anomalies over the Niño 3.4 region (5°S–5°N, 170°–120°W) (Fig. 2b). Positive
(negative) Niño 3.4 index indicates a warm (cold) phase when SST in eastern
tropical Pacific is higher (lower) than the climatological mean (1981–2020). The
40 years can be divided into the warm and cold phases of the JJA SST
anomalies over the eastern tropical Pacific according to Niño 3.4 index.

QBO and Niño 3.4 indices calculated in this study using MERRA-2

reanalysis are highly correlated with those derived from HadISST1 and
NCEP/NCAR reanalysis, with correlation coefficients of 0.98 and 0.97,
respectively. It suggests that the QBO and the eastern tropical Pacific SST
anomaly are well represented in the GEOS-Chem simulations, which is
important for appropriately quantifying impacts of QBO on the interannual
variations in $O_3$ variations over China.

**2.3 Satellite data**

The monthly mean tropospheric column $O_3$ (TCO) data from Ozone

Monitoring Instrument/Microwave Limb Sounder (OMI/MLS) on board the Aura
satellite since 2004 are used to verify the modeled impact of QBO on $O_3$
pollution in China. The grid resolution of OMI/MLS data is 1.25° longitude × 1.0°
latitude, covering the measurement area between 60°S and 60°N. TCO is
calculated by subtracting MLS stratospheric column $O_3$ from OMI total column
$O_3$ (Ziemke et al., 2011). The tropopause height is calculated according to 2 K
km$^{-1}$ lapse rate, which generally locates around 150 hPa in mid-latitudes (Jing
et al., 2006; Peiro et al., 2018). In this study, we used 150 hPa as an
approximation of the tropopause level for the calculation of TCO from the model
simulation, although it may lead to a small bias in the magnitude of TCO.

**3. Result**

**3.1. Impact of QBO on tropospheric $O_3$ in China**

To illustrate the effects of QBO on summertime near-surface $O_3$ over China,

the spatial distribution of the correlation coefficients between the JJA $O_3$





concentrations and concurrent QBO index is presented in Fig. 3a. It shows that the correlation coefficients between QBO index and surface $O_3$ are insignificant over most regions of China, except for part of Qinghai province, which means that the single impact of QBO events cannot significantly affect $O_3$ pollution in China. Previous studies have shown that the impact of QBO can be compounded with ENSO (Sun et al., 2019; Xue et al., 2015). Motivated by these studies, we further examine the relationships between QBO and summertime $O_3$ in the warm/cold phases of SST anomalies of the eastern tropical Pacific. Note that the correlation coefficient between QBO index and Niño 3.4 index is only 0.09, indicating that there is no direct linear relationship between QBO and ENSO, which has also been reported in previous studies (Christiansen et al., 2016; Sun et al., 2019).

The influences of QBO on $O_3$ under different SST anomalies over the eastern tropical Pacific are quite different (Fig. 3b and 3c). During years under the warm SST phase, significant correlations between JJA near-surface $O_3$ concentrations and QBO index are located over the latitudinal band of 25°–40°N in China. In central China (92.5°–112.5°E, 26°–38°N), the correlation coefficient between the regionally averaged $O_3$ concentration and QBO index under the warm phase is 0.53, which is much higher than 0.23 during the whole 40-year period. However, under the cold ENSO phase, there is no significant correlation over China, with a regional correlation coefficient of –0.06. These results suggest that QBO may have a remarkable effect on tropospheric $O_3$ over central China during the warm anomaly of the eastern tropical Pacific SST, while it has little impact on $O_3$ in China during years with cold SST anomalies. Once there is a coincidence of QBOW and warm SST anomaly in eastern tropical Pacific, the combined effects could worsen the $O_3$ pollution over China. Therefore, the three strongest QBOW (1990, 1997 and 2019) and QBOE (1994, 2012 and 2018) years under the warm phase of SST anomaly during the past four decades are chosen to further quantify the influence of QBO on $O_3$ pollution





in China.
Figure 4 presents JJA $O_3$ anomalies in the selected QBOW and QBOE
years relative to the climatological mean (1981–2020). Under the combined
influence of QBOW and warm SST anomaly, positive $O_3$ concentration
anomalies are observed over central and southern China. In contrast, the
surface $O_3$ concentration increases over southern China while it decreases in
central China during QBOE years. The increases in $O_3$ levels over southern
China under warm SST anomaly in both QBOW and QBOE years are due to
the positive correlation between Niño 3.4 index and tropospheric $O_3$
concentrations in southern China. Previous studies have reported that $O_3$
concentrations increased over southern China during El Niño years, which is
related to $O_3$ convergence due to weakened southerlies (Yang et al., 2022; Li
et al., 2022). The different characteristics of $O_3$ changes in central China
highlight the role of QBO in affecting the distribution of $O_3$ over China under
warm SST anomalies of the eastern tropical Pacific.
Figure 5 presents the spatial distribution near the surface and pressure–
longitude cross-sections of absolute and percentage differences between
QBOW and QBOE in $O_3$ concentrations over China under the warm SST
anomaly. Compared with QBOE years, positive $O_3$ concentration anomalies are
located between 25°N and 40°N over China during QBOW, especially over
central China where the maximum anomaly exceeds 3 ppb (parts per billion)
(or 5% relative to the climatological average). The simulated $O_3$ pollution
enhancement is also shown in the vertical distribution of the zonal mean (26°–
38°N) composite differences (Fig. 5b, d). For QBOW years, increased $O_3$
occurred in the whole troposphere, with the maximum increase of 2–3 ppb (3–
5%) between 850 hPa and 500 hPa over central China, indicating a high
probability of enhanced $O_3$ pollution during QBOW relative to QBOE. $O_3$
concentrations also increase in the coastal area of eastern China, which is
mainly due to the decreases in $O_3$ concentrations in the selected QBOE years





relative to the climatological mean, as the $O_3$ concentrations only slightly
increase in the QBOW years. The correlation between $O_3$ and QBO index over
this region is not as strong as that over central China, which indicates that the
anomalous increase in $O_3$ over the coastal area of eastern China may not be a
typical feature of the QBO impact and will not be discussed hereafter.
The modeled difference in summertime tropospheric $O_3$ between the
QBOW and QBOE years can also be observed from satellite (Fig. 6). The
OMI/MLS retrieved TCO are higher in QBOW than QBOE years between 25°–
35°N in China, which is in accordance with the model results. Averaged over
central China, the difference in TCO between the selected QBOW (2019) and
QBOE years (2012 and 2018) from satellite data is 2.8 DU, similar to the 2.5
DU from model simulation. Both model simulations and satellite retrievals
suggest that the QBO can significantly influence tropospheric $O_3$ in China.
**3.2. Mechanism of the QBO impacts on $O_3$ in China**
Composite differences of relevant meteorological variables between the
selected QBOW and QBOE years are shown in Fig. 7 to illustrate mechanisms
of the QBO impacts on $O_3$ in China. During QBOW years under warm SST
anomaly, the decrease in cloud fraction (Fig. 7g) allows more solar radiation to
reach the surface (Fig. 7h) and the RH also decreases over central China (Fig.
7e), relative to QBOE years. These changes in meteorological parameters tend
to increase the photochemical production of $O_3$. However, the air temperature
significantly decreases in the lower (Fig. 7i) and mid-troposphere (Fig. 7f) in
QBOW years compared to QBOE years, which suppresses the $O_3$ production.
The combined effect of the changes in these meteorological parameters leads
to a reduction in net $O_3$ chemical production by about 1% over central China in
QBOW compared to QBOE years based on an integrated process rate analysis
(Lou et al., 2015; Qu et al., 2021; Zhu et al., 2021). Therefore, the chemical
production change is not the major process causing the $O_3$ pollution
deterioration during QBOW years under the warm SST anomaly.





Compared with QBOE years, anomalous northwesterly winds at 850 hPa
occurred over central China during the QBOW years, located at the east edge
of an anomalous high over western China (Fig. 7a). Under the influence of this
anomalous high, the anomalous downdraft over central China (Fig. 7c) can
reduce the vertical transport of $O_3$ to the upper troposphere, which leads to an
$O_3$ accumulation in the lower and mid-troposphere. In addition, the increase in
planetary boundary layer height (Fig. 7d) also favors the vertical $O_3$ mixing
between the lower and upper troposphere in QBOW relative to QBOE years.
Considering the effect of winds on $O_3$ transport, the horizontal JJA $O_3$ mass
fluxes from the surface to 850 hPa and the vertical mass flux at 850 hPa over
central China are calculated and summarized in Table 1. Due to an anomalous
northwesterly, the outflow transport of $O_3$ from the north boundary of central
China is reduced by 1.11 Tg during QBOW years relative to QBOE years.
However, through the east boundary of central China, an inflow transport of $O_3$
is reduced by 1.35 Tg, which overwhelms the gain from the reduced northward
transport. The $O_3$ flux changes through the west and south boundaries are
relatively small and almost offset each other. The overall changes in the
horizontal transport result in a decrease in $O_3$ mass by 0.29 Tg from surface to
850 hPa in QBOW relative to QBOE years, suggesting that the horizontal
advection change is also not the primary process causing the enhanced $O_3$
pollution.
The anomalous downdraft over central China weakens the upward mixing
of high lower-tropospheric $O_3$ concentrations and causes an anomalous
downward transport of $O_3$ by 0.59 Tg at 850 hPa, contributing to the increase
in surface $O_3$ concentrations. Therefore, the impact of the QBO under warm
SST anomaly on the distribution of tropospheric $O_3$ over central China is mainly
via changes in the vertical motion.

### 3.3. Role of China domestic anthropogenic emission

Comparison of the $O_3$ anomaly in BASE and NO_CHN identifies the impact





of China domestic emissions on O₃ concentrations. When domestic
anthropogenic emissions of O₃ precursors are turned off, JJA mean near-
surface O₃ concentrations largely increase across China, especially between
30°–40°N, with maximum increases exceeding 5 ppb during QBOW compared
to QBOE years (Fig. 8a). Averaged over central China, the anomalous increase
in near-surface O₃ concentration is 3.0 ppb in NO_CHN, even higher than that
(1.7 ppb) in BASE simulation. It is consistent with the finding that the vertical
transport plays a dominant role in enhancing the surface O₃ levels over central
China during QBOW years, even though the reduced O₃ photochemical
production, primarily determined by the domestic emissions, weakens the O₃
pollution in QBOW relative to QBOE years in BASE simulation.

Figure 8b shows the simulated vertical distribution of O₃ concentration

difference between the selected QBOW and QBOE years from the NO_CHN
experiment. The positive O₃ anomaly in the troposphere is similar to that from
the BASE experiment, but the increases are mainly between 95°E and 115°E
from the surface to 500 hPa over central China. These results suggest that the
vertical transport process dominates the increase in summertime tropospheric
O₃ concentrations over central China during QBOW under warm SST anomaly
of the eastern tropical Pacific. The reduced photochemical production of O₃
from China domestic anthropogenic emissions is not as important as changes
in the vertical transport in inducing O₃ pollution in QBOW compared to QBOE
years.

**4. Conclusion and discussion**

Based on GEOS-Chem model simulations over 1981–2020, we

investigate the impacts of different QBO events on the surface and tropospheric
O₃ over China. Although only weak correlations are found between JJA mean
near-surface O₃ concentrations and QBO index over China, their positive
correlation is significant in years with warm SST anomalies over the eastern
tropical Pacific. Averaged over central China (92.5°–112.5°E, 26°–38°N), the



correlation coefficient between the regional near-surface $O_3$ concentration and
QBO index during the warm ENSO phase is 0.53. It suggests that the co-
occurrence of the westerly phase of QBO and warm SST anomalies over the
eastern tropical Pacific would exacerbate summertime $O_3$ pollution in China.
Compared with QBOE years, near-surface $O_3$ concentrations increase by up to
3 ppb (5% relative to the average) across China during QBOW, especially over
central China, and the increase in $O_3$ extends from the surface to the upper
troposphere, especially between 850 hPa and 500 hPa.

A combined effect of changes in meteorological conditions (i.e., less cloud,

higher RH, and lower temperature) leads to a slightly lower net $O_3$ chemical
production rate in QBOW years than in QBOE years. Central China is
influenced by anomalous northwesterlies during QBOW, which weakens $O_3$
import from the east boundary and the export from north boundary of central
China, leading to a net $O_3$ export of 0.29 Tg during QBOW, compared to QBOE
years, from surface to 850 hPa. However, change in the vertical transport is the
main process causing $O_3$ concentration increases in QBOW years. An
anomalous downdraft leads to the $O_3$ mass increase of 0.59 Tg below 850 hPa
by suppressing vertical mixing and promoting $O_3$ accumulation in the lower
troposphere. The sensitivity experiment with China domestic anthropogenic
emissions of $O_3$ precursors turned off shows a greater increase of $O_3$ (3.0 ppb)
than that in the default simulation (1.7 ppb). It indicates that the $O_3$ increase
over central China during QBOW years under the warm SST anomaly is mainly
due to the anomalous vertical transport, while a decrease in local chemical
production partly offsets the $O_3$ increases in central China. Moreover, the
positive anomaly of TCO based on GEOS-Chem model simulation is consistent
with the satellite retrieval from the OMI/MLS.

This study explores the effect of QBO on tropospheric $O_3$ over China and

the underlying mechanisms during the warm SST anomalies of the eastern
tropical Pacific, which can improve the understanding of causes of $O_3$ pollution



over China. For climatological average, prevailing easterly winds at 30 hPa
dominate the equator, accompanied by the upward motion over central China
within the troposphere. During QBOW years, the prevailing winds reverse to
westerlies, which may induce the anomalous downward motion over central
China. However, the dynamical mechanism of how the stratospheric QBO
drives changes in the vertical motion and circulation patterns in China along
with the SST anomaly over the eastern tropical Pacific is out of the scope of
this study and merits further investigation. Nevertheless, the QBO index is
positively correlated with the vertical velocity throughout the troposphere over
China, especially between 100°E and 110°E (Fig. 9), where the lower
tropospheric $O_3$ increases the most in the NO_CHN experiment during QBOW
years under the warm SST anomaly. These positive correlations demonstrate
that the weakened (strengthened) upward motion increase (decrease)
tropospheric $O_3$ concentrations during QBOW (QBOE) years, confirming that
changes in the vertical transport driven by QBO events play an important role
in modulating summertime $O_3$ pollution over China. The phenomenon of
changes in tropospheric $O_3$ between different QBO phases is also verified by
satellite retrievals.


***Author contributions.*** YY designed the research; ML performed simulations
and analyzed the data. All authors including HW, LH, PW, and HL discussed
the results and wrote the paper.

***Code and data availability.*** The GEOS-Chem model is available at
https://zenodo.org/record/3974569#.YTD81NMzagR (last access: 1 July 2022).
MERRA-2 reanalysis data can be downloaded at
https://gmao.gsfc.nasa.gov/reanalysis/MERRA-2/ (last access: 1 July 2022).
The monthly mean tropospheric $O_3$ data from OMI/MLS is downloaded from
https://acd-ext.gsfc.nasa.gov/Data_services/cloud_slice/new_data.html (last
access: 1 July 2022). Our model results are available at
https://doi.org/10.5281/zenodo.6793180. $O_3$ observations are obtained from
China National Environmental Monitoring Centre (CNEMC,
http://www.cnemc.cn/en/).

***Acknowledgments.*** HW acknowledges the support by the U.S. Department of
Energy (DOE), Office of Science, Office of Biological and Environmental
Research (BER), as part of the Earth and Environmental System Modeling
program. The Pacific Northwest National Laboratory (PNNL) is operated for
DOE by the Battelle Memorial Institute under contract DE-AC05-76RLO1830.

***Financial support.*** This study was supported by the National Natural Science
Foundation of China (grant 41975159) the National Key Research and
Development Program of China (grant 2020YFA0607803 and
2019YFA0606800) and Jiangsu Science Fund for Distinguished Young
Scholars (grant BK20211541).

***Competing interests.*** The authors declare that they have no conflict of interest.





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





**Table 1.** The horizontal mass flux (Tg) of JJA $O_3$ from the surface to 850 hPa
and the vertical mass flux (Tg) at 850 hPa over central China (92.5°–112.5°E,
26°–38°N). The values are averaged over the selected three QBOW years
(1994, 2012 and 2018) and QBOE years (1990, 1997 and 2019) and their
differences (QBOW-QBOE). Positive values indicate incoming fluxes and
negative values indicate outgoing fluxes.

| | QBOW | QBOE | Difference |
|---|---|---|---|
| | Horizontal mass flux | | |
| East | 1.46 | 2.81 | -1.35 |
| West | 0.92 | 0.74 | 0.18 |
| North | -0.06 | -1.17 | 1.11 |
| South | 3.60 | 3.83 | -0.23 |
| | Vertical mass flux | | |
| Top | -5.68 | -6.27 | 0.59 |





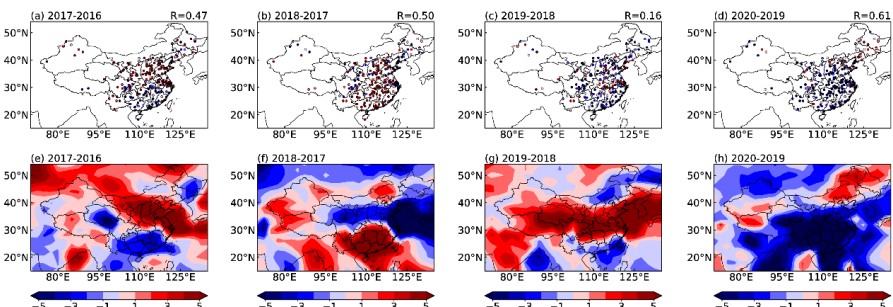

**Figure 1.** Spatial distributions of year-by-year changes in the (a-d) observed and (e-h) modeled JJA $O_3$ concentrations (ppbv) during 2016–2020. The $O_3$ observations are obtained from the China National Environmental Monitoring Centre (CNEMC). Spatial correlation coefficients between simulations and observations are shown at the top right corner of panel a-d.



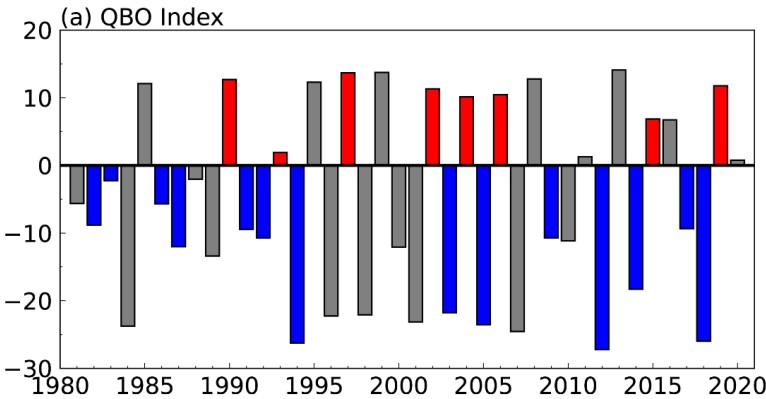

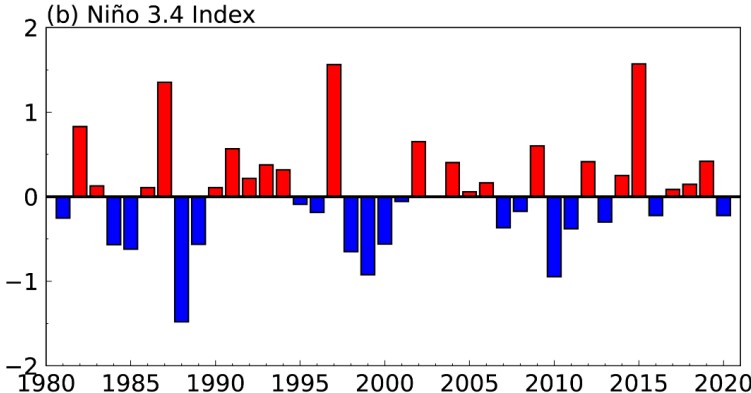

681

682

**Figure 2.** Time series of the JJA mean (a) QBO index (m s$^{-1}$) and (b) Niño 3.4
index (℃) over 1981–2020. Warm phase of SST anomalies over the eastern
tropical Pacific includes 22 years (1982, 1983,1986, 1987, 1990, 1991, 1992,
1993, 1994, 1997, 2002, 2003, 2004, 2005, 2006, 2009, 2012, 2014, 2015,
2017, 2018, 2019) and cold phase includes 18 years (1981, 1984, 1985,
1988, 1989, 1995, 1996, 1998, 1999, 2000, 2001, 2007, 2008, 2010, 2011,
2013, 2016, 2020). Colored bars in (a) indicate years with Niño 3.4 index
above zero.

691

692



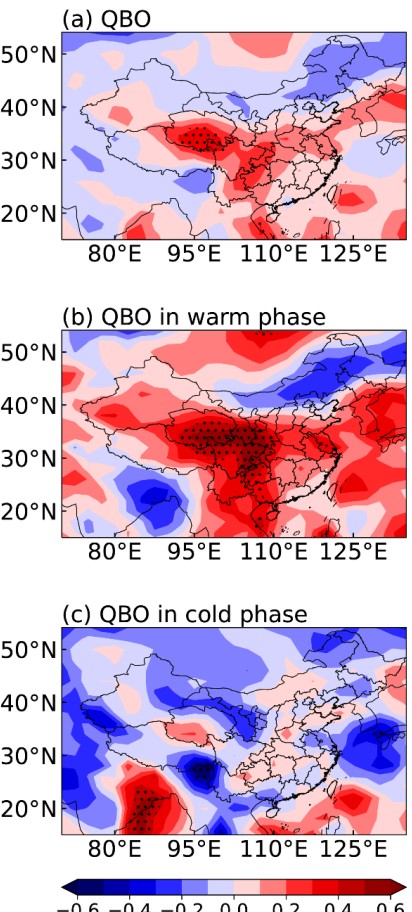

**Figure 3.** (a) Spatial distribution of the correlation coefficients between JJA near-surface $O_3$ concentrations and QBO index over 1981–2020. (b) and (c) are the same as (a), but during years having positive (22 years) and negative (18 years) SST anomalies over the eastern tropical Pacific, respectively. The stippled areas indicate statistical significance at the 90% confidence level.



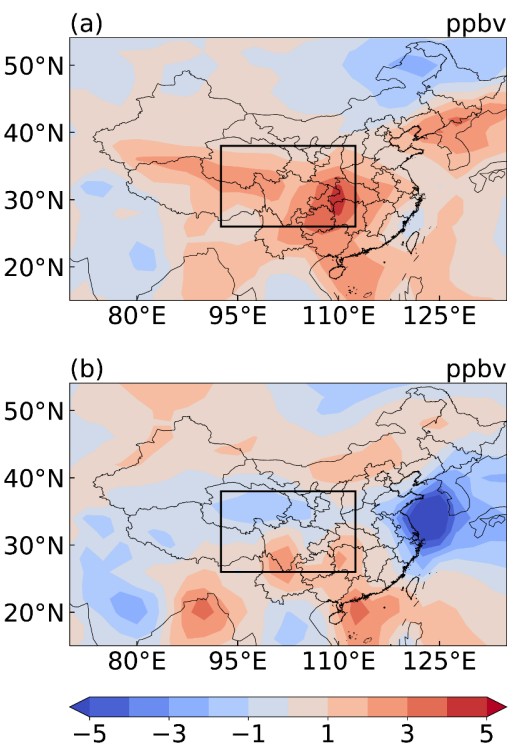

**Figure 4.** Spatial distribution of JJA surface $O_3$ concentration anomalies of (a) the selected three QBOW years (1990, 1997 and 2019), (b) the selected three QBOE years (1994, 2012 and 2018), respectively, relative to the climatological average (1981–2020).

707

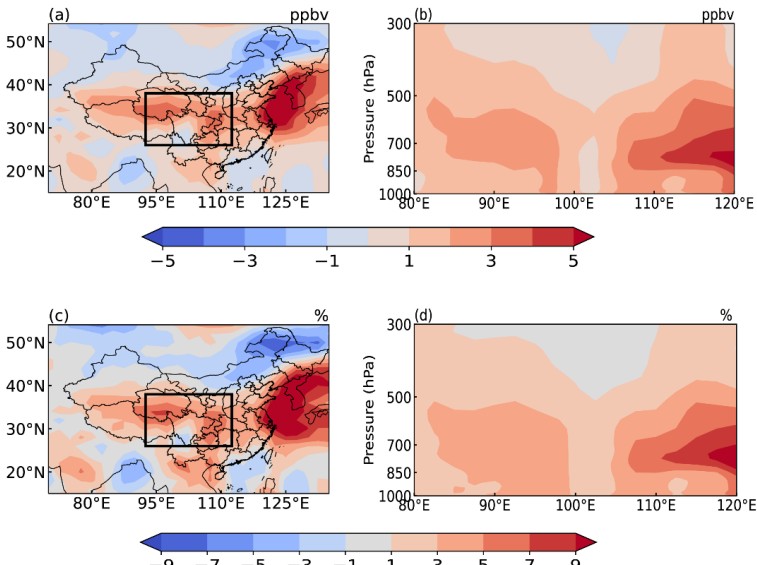

708

**Figure 5.** Spatial distributions of (a) absolute (ppbv) and (c) percentage (%)
differences relative to the climatological mean (1981–2020) in JJA near-
surface $O_3$ concentrations between the selected three QBOW years (1990,
1997 and 2019) and QBOE years (1994, 2012 and 2018) (QBOW–QBOE).
The pressure–longitude cross sections averaged over the latitudes of 26°–
38°N show (b) absolute (ppbv) and (d) percentage (%) differences relative to
the climatological mean in JJA $O_3$ concentrations between the selected three
QBOW years and QBOE years.

717

718

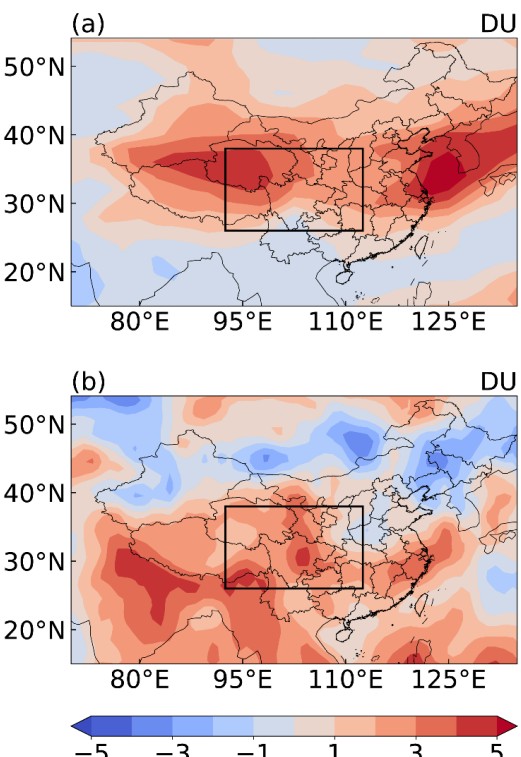

**Figure 6.** Spatial distribution of JJA tropospheric column $O_3$ (TCO, DU) difference between the selected QBOW year (2019) and QBOE year (2012, 2018) based on (a) GEOS-Chem simulations and (b) Aura OMI/MLS.



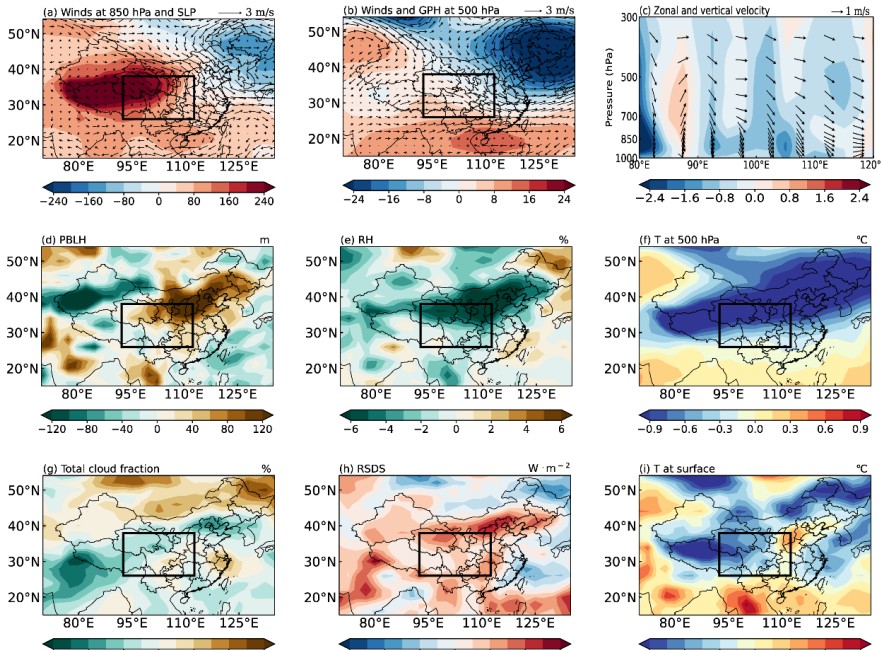

**Figure 7.** Composite differences in the spatial distribution of JJA mean (a) wind fields (m s⁻¹, vector) at 850 hPa and sea level pressure (SLP, Pa, contour), (b) wind fields (m s⁻¹, vector) and geopotential height (GPH, m, contour) at 500 hPa, (d) planetary boundary layer height (PBLH, m), (e) relative humidity (RH, %) at the surface, (f) air temperature (T, ℃) at 500 hPa, (g) total cloud fraction (%),(h) downwelling shortwave radiation at the surface (RSDS, W m⁻²), and (i) surface air temperature (T, ℃) between three QBOW years (1990, 1997 and 2019) and QBOE years (1994, 2012 and 2018) (QBOW–QBOE). **In** (c) the differences in JJA mean zonal wind (m s⁻¹, vector) and vertical velocity (OMEGA, Pa s⁻¹, vector and contour) multiplied by a factor of –100, averaged over 26°–38°N between three QBOW years (1990, 1997 and 2019) and QBOE years (1994, 2012 and 2018) (QBOW–QBOE). The solid black boxes mark central China (92.5°–112.5°E, 26°–38°N).



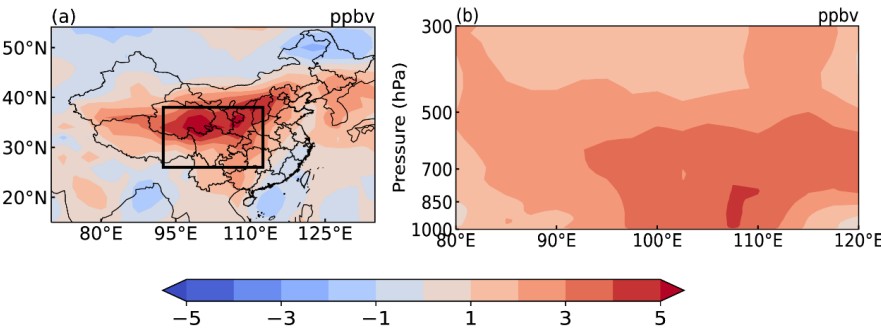

741

**Figure 8.** (a) Spatial distribution of differences in JJA near-surface O₃
concentrations (ppbv) and, (b) the pressure–longitude cross sections
averaged over the latitudes of 26°–38°N of differences in JJA O₃
concentrations (ppbv) between three QBOW years (1990, 1997 and 2019)
and QBOE years (1994, 2012 and 2018) (QBOW–QBOE) from the simulation
that has the China anthropogenic emissions of O₃ precursors turned off
(NO_CHN). The solid black box in a marks central China.





750

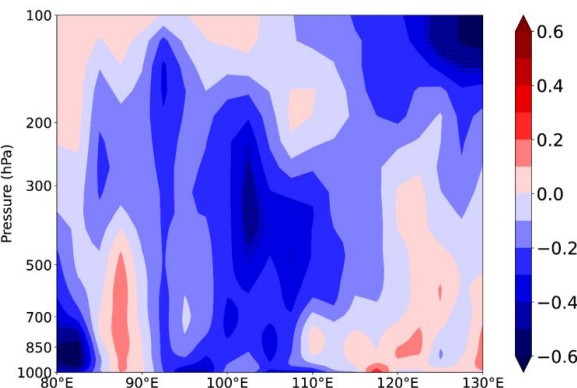

751

**Figure 9.** Pressure-longitude distribution of the correlation coefficients between QBO index and vertical velocity (OMEGA, multiplied by a factor of – 100) in JJA averaged over 26°–38°N for years with warm SST anomaly. The stippled areas indicate statistical significance at the 90% confidence level.