# Peer review of "Ozone pollution in China affected by stratospheric quasi-biennial oscillation"

_Atmospheric Chemistry and Physics, 2022_

## Referee Comment (RC2)

Review of "Ozone pollution in China affected by stratospheric quasi biennial oscillation" by Li et al.

Li et al. proposed a connection between ozone pollution in central China and the stratospheric quasi biennial oscillation (QBO) under the warm phase of the El Niño–Southern Oscillation (ENSO). This topic is interesting and can be potentially important. The authors are commended for their effort to explore the related mechanisms for the proposed connection between QBO and surface ozone in China. The paper reads well. I recommend the paper to be accepted subject to revision that addresses the following points.

L157-159, why? do the emissions change more largely than in other years during the period?

L161-171, Generally, the impact of climatic modes on some processes are investigated using monthly or seasonal climate indices with consideration of different lags, rather than the annual mean.

L258-265, For Figure 6, please explain why the spatial variation of the differences in TCO in (a) and (b) are deferent ?

L276-278, Please elaborate more about this method, how is the 1% calculated?

L287-289, an increase in the boundary height would actually dilute ozone concentrations in the surface. If so, how significant is this process?

L290-308, Table 1 is well done. It suggests that vertical transport is one of the causes for the proposed QBO-ozone relationship. I suggest that the authors provide a complete budget analysis that also includes other components, such as net chemical production and deposition, so it is convincing that vertical transport is the dominant factor.

L326-332, Can these statements be supported by the surface measurements?

Overall, the analysis can be carried out more comprehensively. How can changes in stratospheric wind field in the tropics be connected to surface ozone in the middle latitudes? Are the two climatic modes equally important? or one is more important? The authors can enhance their analysis and thus make their points more convincing in the revision.

Minor points:

Title: this paper is focused on summer season. "summer" needs to be indicated in the title.

L53, replace "it is" to "surface ozone is".

L67, "Yang et al. 2022"? which of "Yang et al. 2022". Please indicate "Yang et al. 2022a" or "Yang et al. 2022b" throughout the manuscript.

L149-152, this may be the case for some regions of China. Ozone pollution can be serious in other seasons in other regions of China.

L181, between  $25^{\circ}$  to  $40^{\circ}$  N, tropopause height decreases with altitudes largely from 100 to-250 hPa. Use a fix 150 hPa can results in some biases.

---

## Author Comment (AC1)

**Manuscript No.: acp-2022-479**

**Responses to Referee #1**

Stratospheric quasi-biennial oscillation (QBO) is an important climate mode that not only modulates the variability in tropical climate system, but also has potential influence globally and may further lead to possible impacts on air quality. Li et al. examined the effects of QBO on interannual variabilities of tropospheric ozone over China mainly by correlation analysis with the help of GEOS-Chem simulations. It is quite an interesting and novel topic. The manuscript is well-organized and easy to follow. I suggest it to be published after addressing my comments below.

We thank the reviewer for all the insightful comments. Below, please see our point-by-point response (in blue) to the specific comments and suggestions and the changes that have been made to the manuscript, in an effort to take into account all the comments raised here.

1. Lines 62-64: The 'downward transport of stratospheric ozone into troposphere' is usually named by 'stratospheric ozone intrusion'. I suggest authors using the proper terminology here.

Response:
    Thank you for the suggestion. We have revised the description as "The intrusion of stratospheric $O_3$ into the troposphere is also one of the sources for near-surface $O_3$ (Zeng et al., 2010; Wespes et al., 2017)."

2. Lines 69: O3 pollution is getting worse in recent years, not decades. At least the two references did not show trends over 10 years.

Response:
    Revised as suggested. "$O_3$ pollution is getting worse in China in recent years (Li et al., 2019; Gao et al., 2022)."

3. Line 153: If the anthropogenic emissions are fixed in 2017, then why use this simulation to evaluate simulated O3 interannual variability? I think changes in anthropogenic emissions could significantly influence O3 year-by-year. Also, I am very confused by the last sentence in this paragraph. Did anthropogenic emissions significantly change between 2018 and 2019?

Response:
    The variations in $O_3$ concentrations are driven by a combination of changes in precursor emissions and the meteorological conditions. Anthropogenic emissions are the largest contributor to variation in tropospheric $O_3$

concentrations over multidecadal timescale, at least for several years (Fu and Tai, 2015; Cooper et al., 2012; Xu et al., 2008). On the interannual time scale, the variations in meteorological conditions have significant influences on surface $O_3$ concentrations (Ding et al., 2019; Li et al., 2020). Therefore, we compared the year-by-year variation in JJA $O_3$ concentrations in observations and BASE simulation rather than the trend, which is more related to the emission changes. The results show that GEOS-Chem could well capture the effect of meteorological parameters on $O_3$ concentrations over China.

The correlation coefficient between the observed and simulated 2018–2019 $O_3$ concentration changes is only 0.16, which may be partly attributed to the emission changes. The Clean Air Action Plan initiated in 2013 rapidly decreased pollutant emissions. However, ozone increased over the 2013–2017 period in the megacity clusters of eastern China (Lu et al., 2020). In 2018, Phase 2 of the Clean Air Action Plan launched, which imposed new emission controls targeted at $O_3$ (Li et al., 2020). This may be one of the reasons for the significant decrease in $O_3$ concentration in 2019 compared to 2018 from observations (Fig. 1c). Ma et al. (2021) also noticed that MDA8 $O_3$ showed a decreasing trend in 2019 relative to 2018, which was opposite to that during 2013–2018.

In addition, Mousavinezhad et al. (2021) suggest that the meteorology in 2019 was favorable to the formation and the accumulation of $O_3$ in BTH, the YRD, and the PRD, by using MLR to separate the contributions from meteorology and precursor emissions to $O_3$ variations. This result is consistent with the increase of $O_3$ in 2019 in BASE simulation, implying a good performance of the model. In this study, the anthropogenic, biomass burning and natural emissions were fixed at 2017 levels to remove the impact of year-to-year emission changes. The spatial correlation coefficient between the observed and simulated $O_3$ concentrations was 0.87 in the year 2017, which also indicated that the GEOS-Chem model is credible in simulating $O_3$ concentration.

We have added a brief description in the manuscript:

The spatial correlation coefficients between the observed and modeled year-by-year changes in $O_3$ concentrations are about 0.5–0.6, except the 2018-to-2019 changes in $O_3$, which could be attributed to the influence of the changes in precursor emissions on the observed $O_3$ concentrations after Phase 2 of the Chinese Clean Air Action Plan launched in 2018 (Li et al., 2020).

4. Lines 150-151: O3 concentrations are accounted during summertime, but QBO and Niño 3.4 indices are calculated with annual climate data (If my understanding is right). Will such mismatch significantly influence the results?

Response:
We apologize for not explicitly describing these indices. QBO and Nino 3.4 indices used in this study are defined as the average of the monthly indices

during June, July, and August. We have added a more detailed interpretation as follows:

The QBO phases are determined by the zonal average of 30 hPa zonal wind over the equator (5°S–5°N) based on MERRA-2 reanalysis (Fig. 2a), with the averages during JJA used in this study.

The Niño 3.4 index averaged over JJA is used to characterize the warm and cold phases of SST anomaly over the eastern tropical Pacific in boreal summer, which is estimated as the SST anomalies over the Niño 3.4 region (5°S–5°N, 170°–120°W).

5.  Lines 174-175: The correlation coefficient numbers should be exhibited in one of the Tables or Figures. The same issue also shows in GEOS-Chem process analysis (lines 277, 294-299 and 305). I strongly suggest authors to add one or several figures to show the differences in O3 budget between QBOW and QBOE.

Response:

We are grateful for the suggestion. The correlation coefficient numbers are added in the top right of each panel of Fig. 2.

The horizontal and vertical $O_3$ mass fluxes are discussed from line 290 to 308, which have been shown in Table 1.

As suggested by the reviewer, we also summarize the process source/sink rates in Table S1. The major processes that influence $O_3$ concentrations include net chemical production, horizontal advection and vertical convention, diffusion and dry deposition. The role of each physical or chemical process can be quantified by the Integrated Process Rate (IPR) analysis. However, we should note that the IPR values represent the instantaneous change in $O_3$ mass, which does not directly reflect the variation of $O_3$ concentrations averaged over a long period.

[Figure]

Figure 2. Time series of the JJA mean (a) QBO index (m s$^{-1}$) and (b) Niño 3.4

index (℃) over 1981–2020. Warm phase of SST anomalies over the eastern tropical Pacific includes 22 years (1982, 1983,1986, 1987, 1990, 1991, 1992, 1993, 1994, 1997, 2002, 2003, 2004, 2005, 2006, 2009, 2012, 2014, 2015, 2017, 2018, 2019) and cold phase includes 18 years (1981, 1984, 1985, 1988, 1989, 1995, 1996, 1998, 1999, 2000, 2001, 2007, 2008, 2010, 2011, 2013, 2016, 2020). Colored bars in (a) indicate years with Niño 3.4 index above zero. The black solid lines represent the indices based on MERRA-2 reanalysis. Bars are QBO index from NCEP/NCEP reanalysis in (a) and Niño 3.4 index from HadISST1 in (b). The correlation coefficients of the indices between MERRA-2 and the NCEP/NCEP reanalysis and between MERRA-2 and HadISST1 are shown in the top right of panels.

**Table S1.** Net rate of change in $O_3$ mass (Tg Season$^{-1}$) of various processes within the planetary boundary layer over central China (92.5–112.5°E, 26–38°N) during the selected three QBOW years (1994, 2012, 2018) and QBOE years (1990, 1997, 2019) and their differences (QBOW-QBOE).

|  | Net chemical production | Horizontal advection | Diffusion and dry deposition | Vertical convection |
|---|---|---|---|---|
| QBOW | 7.42 | -0.35 | -6.42 | 0.43 |
| QBOE | 7.53 | -0.35 | -6.31 | 0.34 |
| Difference | -0.09 | 0.00 | -0.11 | 0.09 |

6. Lines 195-197: Although authors did many analyses in this study to show that O3 differences over China can be significant only when ENSO and QBO were considered together, I still wonder if such insignificant correlations between O3 and QBO were led by some time-lag effects since the tropical QBO signal may need some time to influence China (although I'm not sure about the exact period…). Could authors try some lag-correlation analysis to examine this hypothesis?

Response:

According to the reviewer's suggestion, we have performed the lag-correlation analysis. The results show that the regional correlation coefficient (r) between QBO index and surface $O_3$ in central China is 0.23 (p=0.16) during the whole 40-year period. The lag-correlations between the $O_3$ concentrations over central China and QBO index are even lower, with correlation coefficients of 0.10 (p=0.5) for a three-month of QBO index ahead of the $O_3$ concentrations and –0.14 (p=0.35) for a six-month of QBO index ahead of the $O_3$ concentrations.

We have summarized it in the manuscript as "The lag-correlation analysis is also performed but shows even weaker correlations."

7. Lines 287-289: I don't think changes in boundary layer height could influence vertical O3 transports between lower and upper troposphere. In addition, I suggest authors to clarify the exact levels of the vertical transport. I guess it is mainly in the free troposphere, since 850 hPa -500hPa is higher than boundary layer, but much lower than stratosphere. If so, I believe such downward transport also cannot be considered as a stratospheric ozone intrusion.

Response:
  Many studies suggested that the development of the planetary boundary layer can modulate the vertical extent of turbulent mixing, vertical diffusion and convective transport in the lower troposphere, which affects air pollutant concentrations (Reddy et al., 2012; Gao et al., 2015; Guo et al., 2016; Miao et al., 2018; Gong et al., 2019; Chen et al., 2020; Ma et al., 2021; Duc et al., 2022). Therefore, it is necessary to consider changes in PBLH when analyzing the meteorology impacts on atmospheric pollutants. But we agree with the reviewer that the description about the "between lower and upper troposphere" is incorrect. We have revised it as "In addition, the increase in planetary boundary layer (PBL) height (Fig. 7d) favors the vertical mixing of air within the PBL and the $O_3$-enriched air above the PBL (Gong et al., 2019; Ma et al., 2021)."
  We are grateful for the reviewer's suggestion and have added clarification regarding the levels of the vertical transport (300 hPa) in the manuscript:
  Line 289-292: Under the influence of this anomalous high, the anomalous downdraft throughout the troposphere over central China (Fig. 7c) can reduce the vertical transport of $O_3$ to the upper troposphere, which leads to an $O_3$ accumulation in the lower and mid-troposphere.
  We agree with the reviewer that the stratospheric $O_3$ intrusion may not be the major factor causing $O_3$ increase over central China. Compared with the QBOE years, $O_3$ increases above the surface and up to the upper troposphere, with a maximum increase between 850 and 500 hPa over central China during the QBOW. The increase in $O_3$ is mainly due to the constrained upward mixing of the tropospheric $O_3$ under the influence of the anomalous downdraft from about 300 hPa to the surface. Therefore, we focus on the effect of vertical transport in the troposphere rather than stratospheric ozone intrusion in this manuscript.

8. Lines 317-321: It is interesting that QBO may have higher influences on O3 in China without anthropogenic emission. Could authors add more explanations here? And I am confused by the statement that this finding is consistent with the significant roles of vertical transport. Does the vertical transport become higher in NO_CHN compared to BASE?

[Figure]

Figure A. (a) Spatial distribution of differences in JJA near-surface $O_3$ concentrations (ppbv) between three QBOW years (1990, 1997 and 2019) and QBOE years (1994, 2012 and 2018) (QBOW–QBOE) from the (a) BASE and (b) NO_CHN simulations.

Response:

The increase in $O_3$ near the surface is mainly related to the vertical transport. However, Central China is also affected by an anomalous westerly wind during the QBOW, resulting in a net $O_3$ export of 0.29 Tg from surface to 850 hPa during the QBOW, compared to the QBOE years (Table 1). The net decrease in horizontal transport partly offsets the increase in vertical transport. And the $O_3$ near surface is largely contributed by the domestic anthropogenic emissions. When domestic anthropogenic emissions of $O_3$ precursors are turned off, the net export of $O_3$ horizontal mass flux is only 0.02 Tg (Table S2) and thus the offsetting effect disappears in NO_CHN simulation. Therefore, the $O_3$ increase is stronger in NO_CHN than BASE.

We have provided the mass flux for NO_CHN in Table S2 and revised the description as follows:

Averaged over central China, the anomalous increase in near-surface $O_3$ concentration is 3.0 ppb in NO_CHN, even higher than that (1.7 ppb) in BASE simulation. It results from that the reduction in the net export of horizontal mass flux of $O_3$ due to the removal of domestic emissions (Table S2) leads to a more significant increase in $O_3$ over central China in the NO_CHN experiment.

**Table S2.** The horizontal and vertical mass flux (Tg) of JJA $O_3$ concentration from the surface to 850 hPa over central China (92.5–112.5°E, 26–38°N) based

on NO_CHN simulation. The values are averaged over the selected three QBOW years (1994, 2012, 2018) and QBOE years (1990, 1997, 2019) and their differences (QBOW-QBOE). Positive values indicate incoming fluxes and negative values indicate outgoing fluxes.

|  | QBOW | QBOE | Difference |
|---|---|---|---|
| Horizontal mass flux | | | |
| East | 0.44 | 1.15 | -0.71 |
| West | 0.95 | 0.80 | 0.15 |
| North | 0.77 | 0.09 | 0.68 |
| South | 3.20 | 3.34 | -0.14 |
| Vertical mass flux | | | |
| Top | -4.11 | -4.49 | 0.38 |

9. Lines 329-332: Could authors provide data or numbers to support this statement?

Response:
According to the comment 8, the horizontal transport of $O_3$ is not favorable for $O_3$ accumulation in QBOW compared to QBOE years. When domestic anthropogenic emissions of $O_3$ precursors are turned off, the effect of horizontal transport is weakened, resulting in a more significant increase in the NO_CHN experiment. The original statement is incorrect. We have revised the description in the manuscript:
In the NO_CHN experiment, the reduction in the $O_3$ horizontal export results in a more significant increase of $O_3$ concentration during QBOW compared to QBOE years.

10. Lines 366-373: I suggest to increase some discussion in the dynamic mechanism, although it may slightly beyond the scope of this study. At least one significant question needs to be answered: If the related upward-downward motion transition between QBOE and QBOW are important for O3 in China, why the correlation coefficient between O3 and QBO index is insignificant? What are possible roles of ENSO in influencing meteorological factors in China between QBOW and QBOE years? I believe further discussion depending on data analysis or literature is necessary.

Response:
Thank you for the suggestion. Based on the radiosonde observations, Taguchi (2010) reported a faster QBO downward propagation rate during El Niño conditions, especially westerly QBO phase. Schirber (2015) further analyzed the mechanisms of changes in QBO downward propagation due to

ENSO. Due to the increase in tropospheric temperature under El Niño conditions compared with LA conditions, tropospheric wave activity increases, which strengthens stratospheric QBO forcing. They found that the changes in QBO properties during ENSO were driven by analytical and parametric waves (Christiansen et al., 2016). During El Niño condition, the weaker underlying jet and the increase forcing due to waves cause a faster downward propagation in QBOW. Geller et al. (2016) hypothesized that the more widespread deep convection that occurs in connection with El Niño lead to greater zonally averaged GWMFs, in turn, leading to more rapid descent of QBO westerlies and easterlies. The QBO induced residual circulation propagates downwards, affecting the tropopause and upper troposphere (Zheng et al., 2007). Therefore, we argue that the QBO downlink propagation rate can represent the extent of QBO penetration into the troposphere, implying that a faster propagation rate reflects a more significant impact of QBO on the troposphere. We speculate that this may be the reason why the correlation coefficient between the $O_3$ and QBO indices is insignificant, but shows a significant correlation during El Niño.

Meanwhile, we have added the discussion in the revised version of the manuscript, as follows:

Compared with cold conditions, stratospheric QBO forcing is strengthened due to the increase of tropospheric temperature and changes of analytical and parametric waves under warm SST anomalies of the eastern tropical Pacific, which causes a faster downward propagation in QBO (Taguchi, 2010; Schirber et al., 2015; Geller et al., 2016; Zheng et al., 2007). This may explain why the correlation coefficient between the $O_3$ and QBO indices is insignificant, but shows a significant correlation during warm SST anomalies of the eastern tropical Pacific. The mechanisms deserve further investigation in future studies.

**Reference:**

Chen, L., Zhu, J., Liao, H., Yang, Y., and Yue, X.: Meteorological influences on PM2.5 and O3 trends and associated health burden since China's clean air actions., Sci. Total Environ., 744, 140837, https://doi.org/10.1016/j.scitotenv.2020.140837, 2020.

Christiansen, B., Yang, S., and Madsen, M. S.: Do strong warm ENSO events control the phase of the stratospheric QBO, Geophys. Res. Lett., 43, https://doi.org/10.1002/2016gl070751, 2016.

Cooper, O. R., Gao, R.-S., Tarasick, D., Leblanc, T., and Sweeney, C.: Long-term ozone trends at rural ozone monitoring sites across the United States, 1990–2010, J. Geophys. Res. Atmos. , 117, https://doi.org/10.1029/2012JD018261, 2012.

Duc, H. N., Rahman, M. M., Trieu, T., Azzi, M., Riley, M., Koh, T., Liu, S., Bandara, K., Krishnan, V., Yang, Y., Silver, J., Kirley, M., White, S., Capnerhurst, J., and Kirkwood, J.: Study of Planetary Boundary Layer, Air Pollution, Air Quality Models and Aerosol Transport Using Ceilometers in New South Wales (NSW), Australia, Atmosphere, 13, 176, https://doi.org/10.3390/atmos13020176, 2022.

Ding, D., Xing, J., Wang, S., Chang, X., and Hao, J.: Impacts of emissions and meteorological changes on China's ozone pollution in the warm seasons of 2013 and 2017, Front. Environ. Sci. Eng., 13, 76, https://doi.org/10.1007/s11783-019-1160-1, 2019.

Fu, Y., and A. P. K. Tai, 2015: Impact of climate and land cover changes on tropospheric ozone air quality and public health in East Asia between 1980 and 2010. Atmos. Chem. Phys., 15, 10 093–10 106, https://doi.org/10.5194/acp-15-10093-2015, 2019.

Gao, Y., Zhang, M., Liu, Z., Wang, L., Wang, P., Xia, X., Tao, M., and Zhu, L.: Modeling the feedback between aerosol and meteorological variables in the atmospheric boundary layer during a severe fog–haze event over the North China Plain, Atmos. Chem. Phys., 15, 4279–4295, https://doi.org/10.5194/acp-15-4279-2015, 2015.

Geller, M. A., Zhou, T., and Yuan, W.: The QBO, gravity waves forced by tropical convection, and ENSO, J. Geophys. Res. Atmos., 121, 8886–8895, https://doi.org/10.1002/2015JD024125, 2016.

Gong, C. and Liao, H.: A typical weather pattern for ozone pollution events in

North China, Atmos. Chem. Phys., 19, 13725–13740, https://doi.org/10.5194/acp-19-13725-2019, 2019.

Guo, J., Miao, Y., Zhang, Y., Liu, H., Li, Z., Zhang, W., He, J., Lou, M., Yan, Y., Bian, L., and Zhai, P.: The climatology of planetary boundary layer height in China derived from radiosonde and reanalysis data, Atmos. Chem. Phys., 16, 13309–13319, https://doi.org/10.5194/acp-16-13309-2016, 2016.

Li, K., Jacob, D. J., Liao, H., Shen, L., Zhang, Q., and Bates, K. H.: Anthropogenic drivers of 2013–2017 trends in summer surface ozone in China, Proc. Natl. Acad. Sci. U.S.A., 116, 422–427, https://doi.org/10.1073/pnas.1812168116, 2019.

Li, K., Jacob, D. J., Shen, L., Lu, X., De Smedt, I., and Liao, H.: Increases in surface ozone pollution in China from 2013 to 2019: anthropogenic and meteorological influences, Atmos. Chem. Phys., 20, 11423–11433, https://doi.org/10.5194/acp-20-11423-2020, 2020.

Lu, X., Zhang, L., Wang, X., Gao, M., Li, K., Zhang, Y., Yue, X., and Zhang, Y.: Rapid Increases in Warm-Season Surface Ozone and Resulting Health Impact in China Since 2013, Environ. Sci. Technol. Lett., 7, 240–247, https://doi.org/10.1021/acs.estlett.0c00171, 2020.

Miao, Y., Guo, J., Liu, S., Zhao, C., Li, X., Zhang, G., Wei, W., and Ma, Y.: Impacts of synoptic condition and planetary boundary layer structure on the trans-boundary aerosol transport from Beijing-Tianjin-Hebei region to northeast China, Atmospheric Environ., 181, 1–11, https://doi.org/10.1016/j.atmosenv.2018.03.005, 2018.

Reddy, K. K., Naja, M., Ojha, N., Mahesh, P., and Lal, S.: Influences of the boundary layer evolution on surface ozone variations at a tropical rural site in India, J. Earth Syst. Sci., 121, 911–922, https://doi.org/10.1007/s12040-012-0200-z, 2012.

Schirber, S.: Influence of ENSO on the QBO: Results from an ensemble of idealized simulations, J. Geophys. Res. Atmos., 120, 1109–1122, https://doi.org/10.1002/2014JD022460, 2015.

Taguchi, M.: Observed connection of the stratospheric quasi‐biennial oscillation with El Niño–Southern Oscillation in radiosonde data, J. Geophys. Res., 115, https://doi.org/10.1029/2010jd014325, 2010.

Xu, X., Lin, W., Wang, T., Yan, P., Tang, J., Meng, Z., and Wang, Y.: Long-term trend of surface ozone at a regional background station in eastern China 1991–

2006: enhanced variability, Atmos. Chem. Phys., 13, https://doi.org/10.5194/acp-8-2595-2008, 2008.

Zheng, B., Gu, D., Lin, A., and Li, C.: Dynamical mechanism of the stratospheric quasi-biennial oscillation impact on the South China Sea Summer Monsoon, Sci. China Earth Sci., 50, 1424–1432, https://doi.org/10.1007/s11430-007-0075-z, 2007.

---

## Author Comment (AC2)

**Responses to Referee #2**

Li et al. proposed a connection between ozone pollution in central China and the stratospheric quasi biennial oscillation (QBO) under the warm phase of the El Niño–Southern Oscillation (ENSO). This topic is interesting and can be potentially important. The authors are commended for their effort to explore the related mechanisms for the proposed connection between QBO and surface ozone in China. The paper reads well. I recommend the paper to be accepted subject to revision that addresses the following points.

We thank the reviewer for all the insightful comments. Below, please see our point-by-point response (in blue) to the specific comments and suggestions and the changes that have been made to the manuscript, in an effort to take into account all the comments raised here.

1. Lines 157-159, why? Do the emissions change more largely than in other years during the period?

Response:

The variations in $O_3$ concentrations are driven by a combination of changes in precursor emissions and the meteorological conditions. Anthropogenic emissions are the largest contributor to variation in tropospheric $O_3$ concentrations over multidecadal timescale, at least for several years (Fu and Tai, 2015; Cooper et al., 2012; Xu et al., 2008). On the interannual time scale, the variations in meteorological conditions have significant influences on surface $O_3$ concentrations (Ding et al., 2019; Li et al., 2020). Therefore, we compared the year-by-year variation in JJA $O_3$ concentrations in observations and BASE simulation rather than the trend, which is more related to the emission changes. The results show that GEOS-Chem could well capture the effect of meteorological parameters on $O_3$ concentrations over China.

The correlation coefficient between the observed and simulated 2018–2019 $O_3$ concentration changes is only 0.16, which may be partly attributed to the emission changes. The Clean Air Action Plan initiated in 2013 rapidly decreased pollutant emissions. However, ozone increased over the 2013–2017 period in the megacity clusters of eastern China (Lu et al., 2020). In 2018, Phase 2 of the Clean Air Action Plan was launched, which imposed new emission controls targeted at $O_3$ (Li et al., 2020). This may be one of the reasons for the significant decrease in $O_3$ concentration in 2019 compared to 2018 from observations (Fig. 1c). Ma et al. (2021) also noticed that MDA8 $O_3$ showed a decreasing trend in 2019 relative to 2018, which was opposite to that during 2013–2018.

In addition, Mousavinezhad et al. (2021) suggest that the meteorology in

2019 was favorable to the formation and the accumulation of $O_3$ in BTH, the YRD, and the PRD, by using MLR to separate the contributions from meteorology and precursor emissions to $O_3$ variations. This result is consistent with the increase of $O_3$ in 2019 in BASE simulation, implying a good performance of the model. In this study, the anthropogenic, biomass burning and natural emissions were fixed at 2017 levels to remove the impact of year-to-year emission changes. The spatial correlation coefficient between the observed and simulated $O_3$ concentrations was 0.87 in the year 2017, which also indicated that the GEOS-Chem model is credible in simulating $O_3$ concentration.

We have added a brief description in the manuscript:

The spatial correlation coefficients between the observed and modeled year-by-year changes in $O_3$ concentrations are about 0.5–0.6, except the 2018-to-2019 changes in $O_3$, which could be attributed to the influence of the changes in precursor emissions on the observed $O_3$ concentrations after Phase 2 of the Chinese Clean Air Action Plan launched in 2018 (Li et al., 2020).

2.  Lines 161-171, Generally, the impact of climatic modes on some processes are investigated using monthly or seasonal climate indices with consideration of different lags, rather than the annual mean.

Response:

We apologize for not explicitly describing these indices. QBO and Nino 3.4 indices used in this study are defined as the average of the monthly indices during June, July, and August. We have added a more detailed interpretation as follows:

The QBO phases are determined by the zonal average of 30 hPa zonal wind over the equator (5°S–5°N) based on MERRA-2 reanalysis (Fig. 2a), with the averages during JJA used in this study.

The Niño 3.4 index averaged over JJA is used to characterize the warm and cold phases of SST anomaly over the eastern tropical Pacific in boreal summer, which is estimated as the SST anomalies over the Niño 3.4 region (5°S–5°N, 170°–120°W).

We have also performed the lag-correlation analysis. The results show that the regional correlation coefficient (r) between QBO index and surface $O_3$ in central China is 0.23 (p=0.16) during the whole 40-year period. The lag-correlations between the $O_3$ concentrations over central China and QBO index are even lower, with correlation coefficients of 0.10 (p=0.5) for a three-month of QBO index ahead of the $O_3$ concentrations and –0.14 (p=0.35) for a six-month of QBO index ahead of the $O_3$ concentrations.

We have summarized it in the manuscript as "The lag-correlation analysis is also performed but shows even weaker correlations."

3.  Line 258-265, For Figure 6, please explain why the spatial variation of the

differences in TCO in (a) and (b) are different?

Response:
  The differences in JJA TCO between the selected QBOW year and QBOE year are based on the BASE simulation with emissions fixed at 2017 level, which is only influenced by meteorology fields. However, the differences of TCO from Aura OMI/MLS measurements are driven by a combination of emissions and meteorological conditions. This may be the main reason for the difference in spatial variation of TCO.
  In addition, OMI/MLS tropospheric column ozone was determined daily by subtracting co-located MLS stratospheric column ozone (SCO) from OMI total column ozone each day, known as the tropospheric ozone residual method. This approach involved adjusting for calibration differences between the two instruments, which may cause the retrieval errors (Schoeberl et al., 2007; Liu et al., 2010; Ziemke et al., 2014). Meanwhile, MLS measurements are along-track only. A 2-D interpolation scheme is used to fill in data between orbital gaps to establish daily SCO maps (Ziemke et al., 2006). These factors are responsible for the differences between the GEOS-Chem model and satellites.
  We have added the explanation in the manuscript as "However, it is also noted that the spatial variation of the differences in TCO varies between OMI/MLS and model simulation. It is partly because the emissions were fixed at the 2017 levels during model simulations. These potential biases in satellite retrievals also strongly contribute to the different spatial pattern (Schoeberl et al., 2007; Liu et al., 2010; Ziemke et al., 2006, 2014)."

4.  Lines 276-278, Please elaborate more about this method, how is the 1% calculated?

Response:
  Integrated process rate analysis has been widely conducted to assess the contribution of individual chemical or physical processes to the production and distribution of $O_3$ pollution per unit time in the study domain (Lou et al., 2015; Qu et al., 2021; Zhu et al., 2021). We have added this description in the manuscript.
  According to the integrated process rate analysis, the net chemical production of $O_3$ from surface to the PBL is lower by $-0.09$ Gg d$^{-1}$ over central China during QBOW compared to QBOE years (Table S1).
  The calculation process is as follows:
  $0.09 / 7.53 \approx 1\%$

**Table S1.** Net rate of change in $O_3$ mass (Tg Season$^{-1}$) of various processes from surface to the PBL over central China (92.5–112.5°E, 26–38°N) during the selected three QBOW years (1990, 1997, 2019) and QBOE years (1994, 2012, 2018) and their differences (QBOW-QBOE).

|  | Net chemical production | Horizontal advection | Diffusion and dry deposition | Vertical convection |
|---|---|---|---|---|
| QBOW | 7.42 | -0.35 | -6.42 | 0.43 |
| QBOE | 7.53 | -0.35 | -6.31 | 0.34 |
| Difference | -0.09 | 0.00 | -0.11 | 0.09 |

5. Lines 287-289, an increase in the boundary height would actually dilute ozone concentrations in the surface. If so, how significant is this process?

Response:
The development of the planetary boundary layer can modulate the vertical extent of turbulent mixing, vertical diffusion and convective transport in the lower troposphere, which affects air pollutant concentrations (Guo et al., 2016; Duc et al., 2022). The lower PBLH may constrain vertical mixing and lead to the accumulation of air pollutants (Gao et al., 2015). The increased PBLH was conducive to enhance the atmosphere's ability to disperse particulate matters and improve PM2.5 air quality (Miao et al., 2018, Chen et al., 2020).
However, the sources of troposphere $O_3$ are complex, including downward transport of stratospheric ozone, photochemical reaction products of tropospheric nitrogen oxides (NOx) and volatile organic compounds (VOCs), long-range transport of $O_3$, which makes it difficult to study the influence of PBLH on near-surface $O_3$. Ma et al. (2021) showed that PBLH has a significant positive correlation with MDA8 $O_3$ over NCP region. They used numerical simulations with the National Center for Atmospheric Research Master Mechanism model to quantify the PBLH to the change in surface $O_3$ and found that the increase in PBLHs contribution about 18% to the increment in surface $O_3$. Gong et al. (2019) also reported a positive correlation between $O_3$ concentration and PBLH.
We have modified the biased expression in the revised manuscript as the following:
In addition, the increase in planetary boundary layer (PBL) height (Fig. 7d) favors the vertical mixing of air within the PBL and the $O_3$-enriched air above the PBL (Gong et al., 2019; Ma et al., 2021).

6. Lines 290-308, Table 1 is well done. It suggests that vertical transport is one of the causes for the proposed QBO-ozone relationship. I suggest that the authors provide a complete budget analysis that also includes other components, such as net chemical production and deposition, so it is convincing that vertical transport is the dominant factor.

Response:

As suggested by the reviewer, we summarize the process source/sink rates in Table S1. The major processes that influence $O_3$ concentrations include net chemical production, horizontal advection and vertical convention, diffusion and dry deposition. The role of each physical or chemical process can be quantified by the Integrated Process Rate analysis. However, we should note that the IPR values represent the instantaneous change in $O_3$ mass, which does not directly reflect the variation of O3 concentrations averaged over a long period.

7. Lines 326-332, Can these statements be supported by the surface measures?

Response:
Surface measurements can only tell us the increase or decrease in $O_3$ concentrations, but it is hard to identify which physical or chemical process dominates the $O_3$ change. That is also why we use the model to quantify the importance in individual processes. GEOS-Chem model contains the process analysis module which quantifies the contributions of individual physical and chemical processes to $O_3$ change. The major processes that influence $O_3$ concentrations include net chemical production, horizontal advection and vertical convention, diffusion and dry deposition. The role of each physical or chemical process can be quantified by the Integrated Process Rate (IPR) analysis. In this study, we used IPR analysis to identify the dominant role in enhancing the $O_3$ concentrations during QBOW years.

8. Overall, the analysis can be carried out more comprehensively. How can changes in stratospheric wind field in the tropics be connected to surface ozone in the middle latitudes? Are the two climatic modes equally important? Or one is more important? The authors can enhance their analysis and thus make their points more convincing in the revision.

Response:
The increase in near-surface $O_3$ over central China is mainly attributed to the anomalous downdraft in this study. The QBO of zonal winds is a prominent dynamical feature in the equatorial stratosphere. The QBO is driven by waves generated by convection in the troposphere and propagate upward into the middle and upper atmosphere (Schirber et al., 2015). The potential mechanisms of QBO and tropospheric interaction process have been extensively discussed (Giorgetta et al., 1999; Huang et al., 2012; Lee et al., 2019). Huangfu et al. (2021) suggest that the westerlies over the equator and the easterlies over the offequator form a cyclonic band, providing an upwelling force to the tropopause and affecting Pacific Wallker circulation.
According to the reconstructed records, Wang et al. (2021) show that the development of Walker circulation influences Asian Summer Monsoon strength.

Yuan et al. (2008) suggest that an anomalous reversed Wallker circulation leads to descending motion and hence suppressed convection in the western Pacific, which favors a later onset of the South China Sea summer monsoon. These results imply a significant influence of walker circulation on the climate of the mid-latitudes. Hence, we suspect that the vertical motion over China is obviously influenced by anomaly of walker circulation caused by QBO. The hypothetical mechanism should be proved by model sensitivity experiments. Although the physical mechanism for relationship remains elusive, we believe that our findings would be useful for future air pollution prediction and control. Meanwhile, we have added the discussion in the revised version of the manuscript, as follows:

"Also, it is assumed that the vertical motion over China is influenced by anomaly of Walker circulation caused by the QBO (Huangfu et al., 2021). Although the physical mechanism remains elusive, we believe that our findings would be useful for future air pollution prediction and control."

In the previous study, we investigated the impact of ENSO on summertime near-surface $O_3$ concentrations in China. The results show that simulated near-surface $O_3$ concentrations averaged over southern China (97.5–117.5°E, 20–32°N) present a positive correlation with ENSO index, with statistically significant correlation coefficient between $O_3$ and Niño 3.4 index of +0.55. Furthermore, our analysis suggests that the $O_3$ flux convergence associated with weakened southerlies is the primary cause of the increase in $O_3$ over southern China. And the increased O3 during El Nino years is mainly from domestic emissions.

In this study, QBO has a significant positive correlation with near-surface $O_3$ concentrations over central China (92.5°–112.5°E, 26°–38°N) when the sea surface temperature (SST) over the eastern tropical Pacific is warmer than normal, with a correlation coefficient of 0.53, but QBO has no significant effect on $O_3$ under the cold SST anomaly. Moreover, the $O_3$ increase over central China is mainly attributed to the anomalous downward transport of $O_3$ during the westerly phase of QBO when a warm SST anomaly occurs in the eastern tropical Pacific. When domestic anthropogenic emissions of $O_3$ precursors are turned off, JJA near-surface $O_3$ concentrations largely still increase across China.

Therefore, we find that ENSO has an impact on the near-surface $O_3$ over southern China due to weakened southerlies. However, QBO manifests its impacts on the $O_3$ over central China only during the warm SST phase, which is attributed to the anomalous downward transport of $O_3$. Because of the different regional and dominant mechanisms affected by QBO and ENSO over China, an ad hoc estimate of which modes are more important between ENSO and QBO is difficult. We assume that ENSO and QBO play a synergistic role in modulating $O_3$ pollution in China.

**Minor points:**

9.  Title: this paper is focused on summer season. "summer" needs to be indicated in the title.

Response:
     Thank you for the suggestion. We have changed the title to "Summertime ozone pollution in China affected by stratospheric quasi-biennial oscillation".

10. L53, replace "it is" to "surface ozone is".

Response:
     Changed.

11. L67, "Yang et al. 2022"? which of "Yang et al. 2022". Please indicate "Yang et al. 2022a" or "Yang et al., 2022b" throughout the manuscript.

Response:
     Clarified.

12. L149-152, this may be the case for some regions of China. Ozone pollution can be serious in other regions of China.

Response:
     Thank you for your comment. We have corrected it to "Considering that $O_3$ pollution is most critical during the boreal summer in many regions of China, only summer months (June-July-August, JJA) are examined in this study."

13. L181, between 25° to 40°N, tropopause height decreases with altitudes largely from 100 to-250 hPa. Use a fix 150hPa can results in some biases.

Response:
     Based on previous studies (Jing et al., 2006; Peiro et al., 2018), we used 150 hPa as an approximation of the tropopause level, which results in some biases in the calculation of TCO from the model simulation.
     Meanwhile, we calculated the TCO using 100 and 250 hPa as approximate values of the tropopause in the following Figure. The difference in TCO between the selected QBOW (2019) and QBOE years (2012 and 2018) over central China is 2.7 and 2.2 DU, respectively, which is similar to the 2.5 DU around 150 hPa. Averaged over central China, the difference in TCO from satellite data is 2.8 DU. By comparison, we find that there is a slight bias in the values of the TCO calculated by choosing different heights as the tropopause, but this does not largely affect the qualitative results.

[Figure]

Figure A. Spatial distribution of JJA tropospheric column O$_3$ (TCO, DU) difference between the selected QBOW year (2019) and QBOE year (2012, 2018) based on Aura OMI/MLS (a) using 100 hPa and (b) 250 hPa as approximate values of the tropopause.

**Reference:**

Chen, L., Zhu, J., Liao, H., Yang, Y., and Yue, X.: Meteorological influences on PM2.5 and O3 trends and associated health burden since China's clean air actions., Sci. Total Environ., 744, 140837, https://doi.org/10.1016/j.scitotenv.2020.140837, 2020.

Cooper, O. R., Gao, R.-S., Tarasick, D., Leblanc, T., and Sweeney, C.: Long-term ozone trends at rural ozone monitoring sites across the United States, 1990–2010, J. Geophys. Res. Atmos., 117, https://doi.org/10.1029/2012JD018261, 2012.

Duc, H. N., Rahman, M. M., Trieu, T., Azzi, M., Riley, M., Koh, T., Liu, S., Bandara, K., Krishnan, V., Yang, Y., Silver, J., Kirley, M., White, S., Capnerhurst, J., and Kirkwood, J.: Study of Planetary Boundary Layer, Air Pollution, Air Quality Models and Aerosol Transport Using Ceilometers in New South Wales (NSW), Australia, Atmosphere, 13, 176, https://doi.org/10.3390/atmos13020176, 2022.

Ding, D., Xing, J., Wang, S., Chang, X., and Hao, J.: Impacts of emissions and meteorological changes on China's ozone pollution in the warm seasons of 2013 and 2017, Front. Environ. Sci. Eng., 13, 76, https://doi.org/10.1007/s11783-019-1160-1, 2019.

Fu, Y., and A. P. K. Tai, 2015: Impact of climate and land cover changes on tropospheric ozone air quality and public health in East Asia between 1980 and 2010. Atmos. Chem. Phys., 15, 10 093–10 106, https://doi.org/10.5194/acp-15-10093-2015, 2019.

Gao, Y., Zhang, M., Liu, Z., Wang, L., Wang, P., Xia, X., Tao, M., and Zhu, L.: Modeling the feedback between aerosol and meteorological variables in the atmospheric boundary layer during a severe fog–haze event over the North China Plain, Atmos. Chem. Phys., 15, 4279–4295, https://doi.org/10.5194/acp-15-4279-2015, 2015.

Giorgetta, M. A., Bengtsson, L., and Arpe, K.: An investigation of QBO signals in the east Asian and Indian monsoon in GCM experiments, Clim. Dyn., 15, 435–450, https://doi.org/10.1007/s003820050292, 1999.

Gong, C. and Liao, H.: A typical weather pattern for ozone pollution events in North China, Atmos. Chem. Phys., 19, 13725–13740, https://doi.org/10.5194/acp-19-13725-2019, 2019.

Guo, J., Miao, Y., Zhang, Y., Liu, H., Li, Z., Zhang, W., He, J., Lou, M., Yan, Y.,

Bian, L., and Zhai, P.: The climatology of planetary boundary layer height in China derived from radiosonde and reanalysis data, Atmos. Chem. Phys., 16, 13309–13319, https://doi.org/10.5194/acp-16-13309-2016, 2016.

Huang, B., Hu, Z.-Z., Kinter, J. L., Wu, Z., and Kumar, A.: Connection of stratospheric QBO with global atmospheric general circulation and tropical SST. Part I: methodology and composite life cycle, Clim. Dyn., 38, 1–23, https://doi.org/10.1007/s00382-011-1250-7, 2012.

Huangfu, J., Tang, Y., Ma, T., Chen, W., and Wang, L.: Influence of the QBO on tropical convection and its impact on tropical cyclone activity over the western North Pacific, Clim. Dyn., 57, 657–669, https://doi.org/10.1007/s00382-021-05731-x, 2021.

Jing, P., Cunnold, D., Choi, Y., and Wang, Y.: Summertime tropospheric ozone columns from Aura OMI/MLS measurements versus regional model results over the United States, Geophys. Res. Lett., 33, L17817, https://doi.org/10.1029/2006GL026473, 2006

Lee, J.-H., Kang, M.-J., and Chun, H.-Y.: Differences in the Tropical Convective Activities at the Opposite Phases of the Quasi-Biennial Oscillation, Asia-Pacific J. Atmos. Sci., 55, 317–336, https://doi.org/10.1007/s13143-018-0096-x, 2019.

Li, K., Jacob, D. J., Shen, L., Lu, X., De Smedt, I., and Liao, H.: Increases in surface ozone pollution in China from 2013 to 2019: anthropogenic and meteorological influences, Atmos. Chem. Phys., 20, 11423–11433, https://doi.org/10.5194/acp-20-11423-2020, 2020.

Liu, X., Bhartia, P. K., Chance, K., Spurr, R. J. D., and Kurosu, T. P.: Ozone profile retrievals from the Ozone Monitoring Instrument, Atmos. Chem. Phys., 17, 2010.

Lou, S., Liao, H., Yang, Y., and Mu, Q.: Simulation of the interannual variations of tropospheric ozone over China: Roles of variations in meteorological parameters and anthropogenic emissions, Atmos. Environ., 122, 839–851, https://doi.org/10.1016/j.atmosenv.2015.08.081, 2015.

Lu, X., Zhang, L., Wang, X., Gao, M., Li, K., Zhang, Y., Yue, X., and Zhang, Y.: Rapid Increases in Warm-Season Surface Ozone and Resulting Health Impact in China Since 2013, Environ. Sci. Technol. Lett., 7, 240–247, https://doi.org/10.1021/acs.estlett.0c00171, 2020.

Ma, X., Huang, J., Zhao, T., Liu, C., Zhao, K., Xing, J., and Xiao, W.: Rapid increase in summer surface ozone over the North China Plain during 2013–

2019: a side effect of particulate matter reduction control?, Atmos. Chem. Phys., 21, 1–16, https://doi.org/10.5194/acp-21-1-2021, 2021.

Miao, Y., Guo, J., Liu, S., Zhao, C., Li, X., Zhang, G., Wei, W., and Ma, Y.: Impacts of synoptic condition and planetary boundary layer structure on the trans-boundary aerosol transport from Beijing-Tianjin-Hebei region to northeast China, Atmos. Environ., 181, 1–11, https://doi.org/10.1016/j.atmosenv.2018.03.005, 2018.

Mousavinezhad, S., Choi, Y., Pouyaei, A., Ghahremanloo, M., and Nelson, D. L.: A comprehensive investigation of surface ozone pollution in China, 2015–2019: Separating the contributions from meteorology and precursor emissions, Atmos. Res., 257, 105599, https://doi.org/10.1016/j.atmosres.2021.105599, 2021.

Qu, K., Wang, X., Yan, Y., Shen, J., Xiao, T., Dong, H., Zeng, L., and Zhang, Y.: A comparative study to reveal the influence of typhoons on the transport, production and accumulation of O3 in the Pearl River Delta, China, Atmos. Chem. Phys., 21, 11593–11612, https://doi.org/10.5194/acp-21-11593-2021, 2021.

Peiro, H., Emili, E., Cariolle, D., Barret, B., and Le Flochmoën, E.: Multi-year assimilation of IASI and MLS ozone retrievals: variability of tropospheric ozone over the tropics in response to ENSO, Atmos. Chem. Phys., 18, 6939–6958, https://doi.org/10.5194/acp-18-6939-2018, 2018.
Schirber, S.: Influence of ENSO on the QBO: Results from an ensemble of idealized simulations, J. Geophys. Res. Atmos., 120, 1109–1122, https://doi.org/10.1002/2014JD022460, 2015.

Schoeberl, M. R., Ziemke, J. R., Bojkov, B., Livesey, N., Duncan, B., Strahan, S., Froidevaux, L., Kulawik, S., Bhartia, P. K., Chandra, S., Levelt, P. F., Witte, J. C., Thompson, A. M., Cuevas, E., Redondas, A., Tarasick, D. W., Davies, J., Bodeker, G., Hansen, G., Johnson, B. J., Oltmans, S. J., Vömel, H., Allaart, M., Kelder, H., Newchurch, M., Godin-Beekmann, S., Ancellet, G., Claude, H., Andersen, S. B., Kyrö, E., Parrondos, M., Yela, M., Zablocki, G., Moore, D., Dier, H., von der Gathen, P., Viatte, P., Stübi, R., Calpini, B., Skrivankova, P., Dorokhov, V., de Backer, H., Schmidlin, F. J., Coetzee, G., Fujiwara, M., Thouret, V., Posny, F., Morris, G., Merrill, J., Leong, C. P., Koenig-Langlo, G., and Joseph, E.: A trajectory-based estimate of the tropospheric ozone column using the residual method, J. Geophys. Res., 112, D24S49, https://doi.org/10.1029/2007JD008773, 2007.

Wang, M., Wang, H., Zhu, Z., Yang, X., Zhang, K., Zhang, Y., Liu, W., Zheng, Z., Zong, Y., and Liu, Z.: Late Miocene-Pliocene Asian summer monsoon

variability linked to both tropical Pacific temperature and Walker Circulation, Earth Planet. Sci. Lett., 561, 116823, https://doi.org/10.1016/j.epsl.2021.116823, 2021.

Xu, X., Lin, W., Wang, T., Yan, P., Tang, J., Meng, Z., and Wang, Y.: Long-term trend of surface ozone at a regional background station in eastern China 1991–2006: enhanced variability, Atmos. Chem. Phys., 13, https://doi.org/10.5194/acp-8-2595-2008, 2008.

Yuan, Y., Zhou, W., Chan, J. C. L., and Li, C.: Impacts of the basin-wide Indian Ocean SSTA on the South China Sea summer monsoon onset, Int. J. Climatol., 28, 1579–1587, https://doi.org/10.1002/joc.1671, 2008.

Zhu, J., Chen, L., Liao, H., Yang, H., Yang, Y., and Yue, X.: Enhanced PM2.5 Decreases and O3 Increases in China During COVID-19 Lockdown by Aerosol-Radiation Feedback, Geophys. Res. Lett., 48, e2020GL090260, https://doi.org/10.1029/2020gl090260, 2021.

Ziemke, J. R., Chandra, S., Duncan, B. N., Froidevaux, L., Bhartia, P. K., Levelt, P. F., and Waters, J. W.: Tropospheric ozone determined from Aura OMI and MLS: Evaluation of measurements and comparison with the Global Modeling Initiative's Chemical Transport Model, J. Geohys. Res., 111, D19303, https://doi.org/10.1029/2006JD007089, 2006.

Ziemke, J. R., Olsen, M. A., Witte, J. C., Douglass, A. R., Strahan, S. E., Wargan, K., Liu, X., Schoeberl, M. R., Yang, K., Kaplan, T. B., Pawson, S., Duncan, B. N., Newman, P. A., Bhartia, P. K., and Heney, M. K.: Assessment and applications of NASA ozone data products derived from Aura OMI/MLS satellite measurements in context of the GMI chemical transport model, J. Geophys. Res. Atmos., 119, 5671–5699, https://doi.org/10.1002/2013JD020914, 2014.

---

## Author Response (AR2)

**Manuscript No.: acp-2022-479**

**Responses to Editor**

Thank you for revising the MS. The referees find that the revision is OK. However, I have a comment on the results presented with respect to the ozone change due to QBO.

You have stated that the difference is about 2-3 ppb. This is very small as compared to the tropospheric ozone values (i.e. 40-150 ppb). Similarly, the change in TCO is just 2-5 DU and most measurements even have their uncertainty larger than this value.

Therefore, is this signal of 2-3 ppb or 2-5 DU is significant? If yes, how? This should be discussed in the paper. It is important as most results presented in the paper is purely based the model simulations.

Response:

Thank you for the suggestion. The standard deviation of near-surface $O_3$ over central China simulated in this study is 1.6 ppb and TCO is 1.4 DU. The change in near-surface $O_3$ concentration exceeds 3 ppb over this region between QBOW and QBOE and the change in TCO is 2.5 DU, which are higher than the standard deviations. It suggests that the changes are significant. We have now added the description in the manuscript.

Note that, in the model simulation, with fixed precursor emissions, the $O_3$ variation is only affected by changes in meteorological fields, which may lead to a relatively small $O_3$ variation compared to the real-world values.